# Modeling the Impact of Management Changes on the Infection Dynamics of Extended-Spectrum Beta-Lactamase-Producing *Escherichia coli* in the Broiler Production

**DOI:** 10.3390/microorganisms10050981

**Published:** 2022-05-07

**Authors:** Evelyne Becker, Guido Correia-Carreira, Michaela Projahn, Annemarie Käsbohrer

**Affiliations:** 1MINT VR-Labs, Berliner Hochschule für Technik, 13353 Berlin, Germany; 2Institute of Pharmacy/LPG, Pharmaceutical Biology, Universität Greifswald, 17489 Greifswald, Germany; 3German Federal Institute for Risk Assessment, 12277 Berlin, Germany; guido.correia-carreira@bfr.bund.de (G.C.-C.); michaela.projahn@bfr.bund.de (M.P.); annemarie.kaesbohrer@bfr.bund.de (A.K.); 4Unit of Veterinary Public Health and Epidemiology, University of Veterinary Medicine Vienna, 1210 Vienna, Austria

**Keywords:** antibiotic resistance, ESBL, broiler, management measures, intervention measures, modeling

## Abstract

Livestock animals, especially poultry, are a known reservoir for extended-spectrum beta-lactamase (ESBL)-producing *Escherichia coli (E. coli)*. They may enter the pen either via positive day-old chicks or via the environment. We developed a mathematical model to illustrate the entry and dissemination of resistant bacteria in a broiler pen during one fattening period in order to investigate the effectiveness of intervention measures on this infection process. Different management measures, such as varying amounts of litter, a slow-growing breed or lower stocking densities, were tested for their effects on broiler colonization. We also calculated the impact of products that may influence the microbiota in the chicks’ digestive tract, such as pre- or probiotics, feed supplements or competitive exclusion products. Our model outcomes show that a contaminated pen or positive chicks at the beginning of the fattening period can infect the entire flock. Increasing the amount of litter and decreasing the stocking density were shown to be effective in our model. Differences in the route of entry were found: if the chicks are already positive, the litter quantity must be increased to at least six times the standard of 1000 g/m^2^, whereas, if the pen is contaminated on the first day, three times the litter quantity is sufficient. A reduced stocking density of 20 kg/m^2^ had a significant effect on the incidence of infection only in a previously contaminated pen. Combinations of two or three measures were effective in both scenarios; similarly, feed additives may be beneficial in reducing the growth rate of ESBL-producing *E. coli*. This model is a valuable tool for evaluating interventions to reduce the transmission and spread of resistant bacteria in broiler houses. However, data are still needed to optimize the model, such as growth rates or survival data of ESBL-producing *E. coli* in different environments.

## 1. Introduction

Antibiotics are important drugs for curing bacterial infections. Resistance to these therapeutics hampers proper treatment. Organisms resistant to antimicrobial agents are found in humans, animals, food, plants and in the environment (the water, soil and air) [1,2,3,4,5]. They can be spread from human to human or from animal to human [6,7,8] or via food of animal origin or the environment [9,10,11,12]. As antimicrobial resistance (AMR) is considered a global threat to human health, the World Health Organization has developed a global action plan on AMR [13]. To combat AMR, all sectors need to take action. As regards exposure through the farm-to-fork continuum, measures on the level of livestock farming may reduce the spread of antibiotic resistance and transmission from animals to humans. A known reservoir of extended-spectrum beta-lactamase producing *E. coli*, resistant against penicillins and cephalosporins of the first to the third generation, is livestock, particularly poultry [14]. Due to the high prevalence of ESBL in broiler farms in Europe [10,15] and in fresh retail chicken meat [16,17], it is necessary to investigate measures against ESBL-producing *E. coli* along the entire poultry production chain. Starting from the hatchery [18,19,20], the fattening farm [21,22,23,24,25], the slaughterhouse [26,27,28,29] and even the manure applied as fertilizer to agricultural land [30,31], studies have been conducted.

Current results indicate that the efficacy of the measures is quite variable. Becker et al. [32] reviewed current knowledge on the effectiveness of three different intervention measures: “cleaning and disinfection”, “feed additives” and “competitive exclusion”. They found studies showing a small effect (“feed additives”) and others with a potentially strong impact (“cleaning and disinfection” and “competitive exclusion”) on the incidence of ESBL/AmpC-producing *E. coli* in poultry. Robé et al. [21] investigated the measures “increased amount of litter in the pen”, “reduced stocking density” and “the use of an alternative broiler breed” in an experimental setting. They measured negligible effects on the magnitude of colonization of the broilers with both strains, the ESBL and the pAmpC-producing *E. coli* to which the birds were exposed.

Mathematical models are valuable tools to comprehend the transmission of resistant organisms [33] or to determine the efficacy of mitigation strategies [34], because they are comparatively inexpensive and can compute many experiments in a short time. For example, Huijbers et al. [35] quantified transmission by using a susceptible–infectious–susceptible (SIS) model. In 2018, Plaza-Rodriguez et al. [36] described the transmission dynamics of ESBL/AmpC *E. coli* along the entire broiler production chain. Correia-Carreira et al. [37] modeled interventions in the slaughterhouse. Other models address the development and spread of AMR in humans, pigs or other animals [38]. To our knowledge, there is currently no mathematical model that simulates the dissemination of resistant *E. coli* in broiler fattening farms with the aim of studying the effectiveness of management practices and other interventions. Therefore, we modeled the ingestion and colonization of chickens with ESBL-producing *E. coli*, as well as its spread in a broiler pen and, finally, measures to reduce this infection process.

## 2. Materials and Methods

### 2.1. Mathematical Model

We developed a probabilistic model for the transmission and spread of ESBL-producing *E. coli* in a pen of a chicken-fattening farm. More precisely, it is a mechanistic model which stochastically calculates the growth of ESBL-producing *E. coli* in the pen and in the gastrointestinal tract of the broilers during a fattening period. The mechanistic model on *Salmonella* in pig-fattening farms [39] and the deterministic bacterial population model from Græsbøll and colleagues on multiple bacterial strains in pigs [40] served as the basis for the development of our model described here.

The model was written in R version 4.1.1 [41]; all data were also analyzed and plotted by using R.

Our model simulates a fictitious pen (one room) with ESBL-positive or negative chickens picking up feed, as well as potentially contaminated bedding material. The bacteria multiply in the gastrointestinal tract and are excreted into the litter. In the model, transmission of the bacteria from chicken to chicken occurs via the litter, i.e., excreted bacteria in the litter are picked up by the chickens.

The following three processes are modeled (see Figure 1) to describe what happens to ESBL-producing *E. coli* during the course of a day:Intake: feed, water and bedding material (including bacteria, N_ing_) are ingested.Growth: ESBL-producing *E. coli* grow in the chick’s intestinal tract (N_int_).Excretion: Intestinal ESBLs were excreted with the feces (N_fec_).

The bacteria, excreted with the feces, accumulate in the litter and spread in the pen (N_lit_). Some of this contaminated bedding material is picked up again by the chicks.

The following general assumptions and parameters apply to our model:All chickens share the same pen. The pen area is calculated for each scenario based on the chosen values of the parameters “total number of chickens”, “target weight” and “stocking density”.The excreted bacteria are immediately mixed evenly in the total amount of available litter. Each chick ingests the same amount of litter and, thus, bacteria.All the bacteria considered are resistant. If there is even a single resistant bacterium in the gastrointestinal tract of a chick, it is considered to be ESBL-positive.Once a viable colony-forming unit enters the intestinal tract, the growth rate, once assigned, does not change for these bacteria in this chick.Infection occurs only via positive chicks, artificial oral inoculation or ingestion of contaminated bedding material; we did not model a transmission via bird-to-bird-contact or air.The amount of excreted feces per chick per day is calculated from the difference between ingested feed and water (minus the amount of exhaled water) and daily weight gain. The loss of feed (energy) required for metabolism was not calculated and subtracted.

### 2.2. Modeling the Processes

#### 2.2.1. Uptake of ESBL-Producing *E. coli*

The chickens entering the pen can be negative or ESBL-positive from a certain day. In addition to the daily feed intake, the exact amounts per day of which were taken from the breeders manual (see Table 1), the material ingested from the litter is estimated to be around 1.4% of the feed intake [42]. If the litter is contaminated, either with ESBL-producing *E. coli* from a previous flock, e.g., due to incomplete cleaning and disinfection, or from feces of currently infected birds, the chickens pick up ESBL-producing *E. coli* together with the bedding material.

Equation (1) describes the ingested amount of colony-forming units (CFU) from the litter by one chick (i) on one day (j) (N_ing_). The amount of feed ingested on that day in grams is taken from the breeder’s manual [44,45]. This value is multiplied with 0.014, which is the amount of litter ingested by the chicks per day, expressed as a fraction of the amount of feed consumed per day (in grams). This factor k_ing_ = 0.014 is derived from Malone et al. [47]. Finally, N_lit_(j) is the amount of colony-forming units in the litter on that specific day, and M_pen_ (j) is the sum of the mass of litter and excrement in the pen on that day.
(1)$$N_{\mathrm{ing}}\left(i,j\right)=\frac{I_{f}\left(j\right)\cdot k_{\mathrm{ing}}\cdot N_{\mathrm{lit}}\left(j\right)}{M_{\mathrm{pen}}\left(j\right)}$$

The calculated number of bacteria present in the intestine from the beginning of the fattening period to day j is given by Equation (2):(2)$$N_{\mathrm{int}}\left(i,j\right)=\{\begin{array}{c} N_{\mathrm{ing}}\left(i,j\right)\cdot k_{\mathrm{via}}\;\;\;\;\;\;\;\;\;\;\;\;\;\;\;\;\;\;\;\;\;\;\;\;\;\;\;\;\;\;\;\;\;\;\;\;\mathrm{for}\;j=1\\ N_{\mathrm{ing}}\left(i,j\right)\cdot k_{\mathrm{via}}+\overset{j-1}{\underset{n=1}{\sum }}\frac{1}{n+1}N_{\mathrm{ing}}\left(i,j-n+1\right)\cdot k_{\mathrm{via}}\;\;\mathrm{for}\;j\geq 2 \end{array}$$
where N_int_ (i,j) is the amount of accumulated CFU in the intestine of chicken i on day j, and k_via_ is the factor describing survival in the gastrointestinal tract. Moreover, k_via_ is a random number between 0 and 0.5, drawn from the uniform distribution U(0, 0.5) as we model survival as a stochastic event.

#### 2.2.2. Growth of ESBL-Producing *E. coli* in the Intestine

The growth of viable bacteria in the intestine is described by Equation (3).
(3)$$N_{\mathrm{intgrowth}}\left(i,j\right)=\left(1-k_{\mathrm{exc}}\right)\cdot N_{\mathrm{int}}\left(i,j\right)\cdot 10^{k_{\mathrm{gro}}}$$
where N_intgrowth_ is the total number of ESBL *E. coli* CFU in the intestine after bacterial growth and loss of bacteria due to fecal excretion. Excretion is described by using the excretion factor, k_exc_, which is the proportion of CFU lost daily due to defecation. Bacterial growth of the remaining amount of ESBL in the intestines is then modeled by multiplying the number of CFU by 10 to the power of a growth factor (k_gro_), which is a random number between 0 and 5, drawn from a uniform distribution U(0,5), as we model growth as a stochastic process.

Growth is limited to 8 log_10_ CFU/g to represent the maximum carrying capacity, as measured in the colon by Robé et al. [49] and Dame-Korevaar et al. [50].

#### 2.2.3. Excretion of ESBL-Producing *E. coli* into the Litter

The amount of excreted feces is calculated as the difference between the total intake (feed and water) and the body weight gain according to Equation (4):(4)$$M_{\mathrm{exc}}=M_{\mathrm{fw}}-M_{\mathrm{bwg}}$$
where $M_{\mathrm{exc}}$ is the mass of excreted material that day in grams; and $M_{\mathrm{fw}}$ is the mass of water and feed taken in that day, but excluding 30% of the water mass, which is assumed to be lost by exhalation. Finally, $M_{\mathrm{bwg}}$ is the mass of body weight gain for that day.

We did not deduct feed for basal metabolic rate, but 30% water for respiration [47]. During the first 24 h, day-old chicks are supplied with nutrients from the yolk sac and do not consume any feed. Therefore, the amount of feces was estimated for the days when the calculated feed intake was 0–10 g.

The chick consumes and excretes the daily amount of feed and water throughout the entire day. In the model, we consider only the total intake and excretion in one day. We deterministically estimated from the literature data that the passage of the feed mash through the gastro-intestinal tract of the chicken takes 4.95 h [51,52,53]. Thus, we assume that the gastrointestinal tract is filled and emptied approximately 4.85 times a day. Accordingly, the total mass of contents in the gastrointestinal tract is estimated to be $M_{\mathrm{exc}}/4.85$.

It is assumed that only a part of the bacteria grown in the intestine gets into the pen by excretion; the other part remains in the chickens, for example, in the caeca. Therefore, the excreted amount of CFU/g feces (N_fec_) is calculated by using the excretion factor, k_exc_ (Equation (5)).
(5)$$N_{\mathrm{fec}}\left(i,j\right)=k_{\mathrm{exc}}\cdot \left(\frac{N_{\mathrm{int}\left(i,j\right)}}{M_{\mathrm{fec}\left(i,j\right)}}\right)$$

The excreted bacteria of the day are added to the CFU already present in the litter (Equation (6)). However, the amount of CFU in the litter from the previous day is reduced by an estimated factor, k_sur_ = 0.5, as we assume that some bacteria do not survive in the bedding material.
(6)$$N_{\mathrm{lit}\left(i,j\right)}=\left(N_{\mathrm{lit}\left(j-1\right)}\cdot k_{\mathrm{sur}}\right)+\underset{i,j}{\sum }N_{\mathrm{fec}}$$

Subsequently, the amount of pen material must be recalculated by adding the newly excreted feces (M_fec_) of all chicks on that day to the existing amount of material (M_pen_) (Equation (7)). The model does not take into account the evaporation of water from the mixture of bedding material and feces.
(7)$$M_{\mathrm{pen}}\left(j\right)=M_{\mathrm{pen}}\left(j-1\right)+\underset{i.j}{\sum }M_{\mathrm{fec}}$$

The final calculation of the contamination of the litter introduced by feces is performed at the end of a simulated day. Thus, at any simulated day, it is the contamination of the previous day which affects the chickens. Therefore, in the model, ingestion of contaminated litter generally begins on day 2, with the exception of the “contaminated pen” scenario, where an initial contamination of the pen allows ingestion of contaminated litter on day 1. Correspondingly, the end of the modeled fattening period includes one more day than defined in the setting, i.e., 22 days, if the fattening period was set to 21 days to include contamination on the last day of the fattening period (day 21) which only affects the animals on the next modeled day (day 22).

### 2.3. Adaptation to Experimental Data

We used published data from Dame-Korevaar et al. [43,50] and Robé et al. [21,49] to verify the model on the basis of experimental data and adjust if necessary. These seeder-bird models provide us with the required precise amounts of bacteria ingested and excreted by the chickens. Using these data, we concretized the values for the variables k_via_ and k_exc_, the factor of bacteria that remain capable of reproduction when ingested and the excretion factor (k_exc_). At the beginning of our computation experiments we used estimated numbers: a range of 0 to 0.6 for k_via_ and the factor 2 for k_exc_.

We modeled experiments with 5 seeders and 5 sentinels (ratio 1:1) in a pen with an area of 1 m^2^ (target weight 2 kg), a feeding duration of 21 days and an inoculation of 10^2^ CFU ESBL-producing *E. coli* per bird on day 1, comparable to Dame-Korevaar et al. [43]. We also modeled experiments of Robé et al. [49] with 4 seeder and 16 sentinel birds (ratio 1:5) in a pen of 4.6 m^2^ (target weight 2 kg, stocking density 39 kg/m^2^) and an inoculation amount of 10^2^ CFU ESBL-producing *E. coli* per chick on day 3.

### 2.4. Modeling Different Ways of Infection

ESBL-producing *E. coli* enter our fictitious pen in only two ways:The day-old chicks are positive for ESBL-producing *E. coli*.The pen is contaminated, and the chickens peck and ingest contaminated litter. Contamination may come from internal (e.g., previous positive flocks) or external sources (e.g., environment and rodents); the origin is not considered in our model.

For the first scenario, we adopted the seeder-bird model and set 18 from 90 chicks (ratio 1:5) positive with an amount of 10^2^ CFU per chick on day 1. The basic setting consisted of 90 Ross 308 chickens, 18 seeders, 72 sentinels, a feeding duration of 36 days, a target weight of 2.332 kg, a stocking density of 39 kg/m^2^ and 1000 g of litter per square meter. The pen area was calculated from the number of chickens, the target weight and the stocking density. Here it is 5.38 m^2^, from which the amount of litter at the beginning of the fattening period was derived, i.e., 5.38 kg. This setting served as a control group with which we subsequently compared the results of the groups where we changed management practices or calculated the impact of other measures.

For the second scenario, we modeled a contaminated pen at the beginning of the fattening period. We tried different amounts of ESBL-producing *E. coli*. In our model, these bacteria were not located in individual places but were evenly distributed in the litter. All chicks were set to be negative at the beginning. We also used the breed Ross 308, a stocking density of 39 kg/m^2^, a target weight of 2.332 kg, 1000 g litter/m^2^ and a feeding duration of 36 days. The pen area was also 5.38 m^2^, and the amount of litter was 5.38 kg. The setting where contamination was just high enough to result in a positive flock throughout the entire fattening period then served as the control group for the calculations where the measures were modeled.

### 2.5. Modeling Changes in Management Practice

Changes could be inserted into the modeled processes at various points, and, thus, management measures could be simulated. We then calculated the impact on the bacterial load of broilers in the pen.

The following measures were tested for their effect on chick colonization for both scenarios, a pen with positive chicks and a contaminated pen.

Amount of litter at the beginning of the fattening period: 3000, 6000 and 9000 g/m^2^, instead of 1000 g/m^2^;Breed Rowan x Ranger with a feeding duration of 47 days, daily feed intake and daily body weight gain according to the corresponding Aviagen manual [45] and a target weight of 1.911 kg on day 47, instead of Ross 308 with a feeding duration of 36 days, a target weight of 2.332 kg and daily intake and daily body weight gain according to the corresponding Aviagen manual [44];Stocking density: 20 or 25 kg/m^2^, instead of 39 kg/m^2^;Different combinations of the previously mentioned measures;Products with impact on the microbiota, such as prebiotics, probiotics, synbiotics, feed additives or competitive exclusion, were expected to reduce the prevalence of ESBL-producing *E. coli* [23,25]. We modeled the effect of these products with a lower maximum growth rate of ESBL-producing *E. coli* in the intestine—10^4^ instead of 10^5^.

### 2.6. Model Runs

Our model was initiated by selecting the total number of chickens, number of seeders and sentinels, the farming conditions, breed, litter amount and stocking density, as well as the initial amount of “inoculated” CFU and the day when the chicks are positive, or the number of CFU in the pen. In each run, 100 flocks were simulated.

The calculated numbers were written in tables with absolute numbers of CFU and recalculated as log_10_ CFU (Appendix A). As described above, we model an additional day beyond the nominal fattening period. Therefore, the tables include an additional day after the end of the fattening period, i.e., day 37 for the 36-day fattening period. Mean values, including standard deviation, were calculated first over all chicks and then over all 100 iterations.

We plotted the mean prevalence of ESBL-positive chickens on each day of the fattening period; the mean number of excreted bacteria, averaged over all animals [CFU/g feces]; the amount of litter plus feces (pen mass) (g); the number of bacteria per square meter in the pen [log_10_ CFU/m^2^]; the number of bacteria per g pen mass [log_10_ CFU/g]; the mean total number of bacteria in the intestine, averaged over all animals [log_10_ CFU]; the mean of total CFU excreted, averaged over all animals; and the mean of total CFU ingested, averaged over all animals.

We used the prevalence, the number of excreted bacteria per gram feces [CFU/g feces] and the number of bacteria per gram pen mass [log_10_ CFU/g] to compare the measures in their effects in both scenarios, positive chicks at the beginning or contaminated pen.

## 3. Results

### 3.1. Adaptation to Experimental Data

The parameters we used to model the processes were extensively reviewed and finally chosen to be as close as possible to the description of events in vivo. However, at the beginning of our modeling experiments, we had to estimate two factors: the maximum number of bacteria that remain capable of reproduction after ingestion (k_via_); and the excretion factor (k_exc_), which describes the fraction of bacteria grown in the gut that is excreted. We then used the empirical seeder-bird experiments of Dame-Korevaar et al. [43,50] and Robé et al. [21,49] to test and optimize the values of these parameters. We found that we needed to set k_via_ to a maximum of 0.5 and k_exc_ to 0.3 in order for our results to best match published data from colleagues, in terms of curve progression and data for mean bacteria excreted, averaged over all animals [log_10_ CFU/g feces] or bacteria per g pen mass [log_10_ CFU/g]. We were then able to show that five seeder chicks set positive with 10^2^ CFU ESBL each on day 1 were sufficient to make the other five sentinel birds positive within a few days (setting: Ross 308 birds, target weight 2 kg, stocking density 20 kg/m^2^, feeding duration 21 days and pen area 1 m^2^). On day 5, i.e., 72 h after the seeder are classified positive in the model (day 2), 97.2 ± 9.13% of all chickens were positive. Subsequently, on day 7 of the fattening period, 99.2 ± 4.42% of the chickens were positive and stayed positive until the end of the fattening period (Figure 2 and Table 2). The excreted amount of CFU/g feces is calculated with a maximum of 4.51 ± 4.42 log_10_ at the end of the fattening period on day 22. In the litter, it is up to 3.71 ± 3.66 log_10_ CFU/g on day 17 (Table 2). If the settings are changed to those of Robé et al. [49], i.e., a seeder–sentinel ratio of 1:5, 20 chickens in total, breed Ross 308, feeding duration 36 days, target weight 2 kg, stocking density 39 kg/m^2^, giving a pen area of 4.6 m^2^, and inoculation on day 3, the results are also similar. Setting the seeder birds positive with 10^2^ CFU per seeder, 87.55 ± 24.96% of the chicks are positive 72 h after inoculation, i.e., on day 7 in our model. The maximal prevalence is reached on day 15 with 97.15 ± 14.67%. This prevalence persists until the end of the fattening period (see Figure 3 and Table 2). The excreted amount of CFU/g feces is calculated with a maximum of 3.6 ± 3.43 log_10_ on day 23. There are up to 4.41 ± 4.26 log_10_ CFU/g in the litter on day 23 (Table 2).

In addition, we tested an inoculation amount of 10^1^ CFU per bird to confirm our model, but this amount did not infect all birds, regardless of the seeder–sentinel ratio (Appendix A).

### 3.2. Modeling Different Ways of Infection

According to our calculation, the seeder-bird model with an inoculation amount of 10^2^ CFU per chick was suitable to infect almost all sentinel birds (ratio 1:5) within a few days (see 3.1). We chose these values for our baseline model to simulate as realistic a scenario as possible in which day-old chicks arrive at the farm partially ESBL-positive. In this context, we also chose day 1 to set the chicks positive, thus modeling a flock of 90 Ross 308 chickens, 18 of which are positive from the beginning, each with 10^2^ CFU of resistant bacteria, i.e., an initial prevalence of 20%, a feeding duration of 36 days, a stocking density of 39 kg/m^2^ and 1000 g litter per square meter (Figure 4). The maximum prevalence was reached on the sixth day of the fattening period, where 99.38 ± 1.98% of the chickens were positive and remained infected until the end of the fattening period (Figure 4 and Table 3). The excreted amount of CFU/g feces was a maximum of 4.63 ± 4.08 log_10_ on day 20, and in the litter, 3.9 ± 3.47 log_10_ CFU/g on day 18 (Table 3). These results serve as a reference with which we compared the results of the groups where we changed management practices or implemented measures.

For the second scenario, “contaminated pen”, we tested different values for the number of bacteria in the litter at the beginning of the fattening period. A contamination of less than 10^6^ CFU in the pen was not sufficient to infect even a single chick (Appendix A); thus, a dissemination could not start. A number of 10^6^ CFU evenly distributed in 5.38 kg litter, i.e., initially 185 CFU/g litter or 2.27 log_10_ CFU/g, was sufficient to initiate the infection process. Almost all chicks were positive on day 7, with a prevalence of 99.79 ± 0.61% (Table 3), and they remained so throughout the rest of the fattening period (Figure 5). The excreted amount of CFU/g feces was a maximum of 4.51 ± 3.85 log_10_ on day 26, and a maximum of 3.68 ± 3.36 log_10_ CFU/g was calculated in the litter for day 20 (Table 3).

### 3.3. Calculated Effects of Management Measures at Farm Level

#### 3.3.1. Increasing the Amount of Litter per Square Meter

In the scenario where some chicks are positive at the beginning, our model calculated a negligible effect on the colonization of the chickens when we increased the litter amount from 1000 to 3000 g/m^2^. A prevalence of 98.53 ± 6.24% was reached on day 11 (Figure 6 and Table 3), the maximum CFU/g feces was 4.19 ± 3.92 log_10_ on day 34 (Figure 7 and Table 3) and the maximum contamination of litter was calculated to be 3.25 ± 3.04 log_10_ CFU/g on day 30 (Figure 8 and Table 3). A decrease is observed when the amount of litter is further increased to 6000 g/m^2^, but the prevalence does not drop below 20% (Figure 6), which is almost the same as the initial prevalence. The highest prevalence of 57.23 ± 25.11% is reached on day 7 (Figure 6 and Table 3), and the amount of CFU/g feces reaches a maximum of 2.63 ± 2.37 log_10_ on day 2 (Figure 7 and Table 3). The litter contains a maximum of 1.13 ± 0.77 log_10_ CFU/g on day 3 (Figure 8 and Table 3). When the amount of litter is increased to 9000 g/m^2^, the transmission and spread of ESBL nearly stops in our modeled pen. The initial prevalence of 20% increases only to a maximum of 29.72 ± 16.97% on day 5, after which the curve drops to 5.29 ± 1.97% on day 37 (Figure 6 and Table 3). The maximum number of bacteria in feces and litter decreased drastically (see Figure 7 and Figure 8 and Table 3).

In the contaminated-pen scenario, the impact was more obvious. Already with an increase of the litter quantity to 3000 g/m^2^, the infection process comes to a standstill (see Figure 6 and Table 3). The contamination of 10^6^ CFU was, at that point evenly distributed in 16.14 kg of the litter, i.e., an initial load of 62 CFU per g litter or 1.79 log_10_ CFU/g. The maximum prevalence on day 3 was only 32.46 ± 4.42%, and the maximum number of bacteria in feces and litter decreased drastically (Figure 7 and Figure 8 and Table 3). When the amount of litter was further increased to 6000 or 9000 g/m^2^, the infection of the chickens did not even begin (see Figure 6 and Table 3), so that neither plottable values for the mean of excreted bacteria, averaged over all animals [log_10_ CFU/g feces] nor for the bacteria per g pen mass [log_10_ CFU/g] are available; thus, at this point, there are also no figures.

#### 3.3.2. A Slow-Growing Breed

With the breed Rowan x Ranger instead of Ross 308, nearly no effect was calculated compared to the reference group in both scenarios when one-fifth of the chickens were set positive or when the pen was set positive at the beginning of the feeding period. The numeric values are almost the same (see Table 3). The prevalence reached 99% at day 6 (positive chicks) or at day 5 (positive pen), and the curves remained at nearly 100% until day 48 (Figure 6). The maximum excreted amount of CFU/g feces was 4.23 ± 4.01 log_10_ on day 25 (positive chicks) and 4.01 ± 3.7 log_10_ on day 30 (positive pen) (see Figure 7). The litter contained a maximum number of bacteria with 3.4 ± 3.33 log_10_ CFU/g on day 21 (positive chicks) and 3.11 ± 2.84 log_10_ CFU/g on day 27 (positive pen) (see Figure 8).

#### 3.3.3. Lower Stocking Densities

When positive day-old chicks entered our model pen, a reduction in the stocking density had little effect on the dissemination of ESBL-producing *E. coli*. A stocking density of 25 or 20 kg/m^2^ showed the same prevalence of 99% on day 7. Moreover, the amounts of excreted CFU and ESBLs in the litter are similar, as well: around 4.5 log_10_ CFU/g feces, and about 3.6 log_10_ CFU/g litter (Figure 6, Figure 7 and Figure 8 and Table 3). Considering the second scenario, where the litter was contaminated at the beginning, there was a minor effect with a stocking density of 25 kg/m^2^ and a strong effect with a stocking density of 20 kg/m^2^. At a stocking density of 25 kg/m^2^, the prevalence curve resembles that of the reference group but does not exceed 90.64 ± 28.71% (Figure 6 and Table 3). The maxima for excreted ESBLs with 4.1 ± 3.98 log_10_ CFU/g and for bacteria in the litter with 3.2 ± 3.04 log_10_ CFU/g were slightly lower and occurred later than in the reference group (Table 3). Choosing a stocking density of 20 kg/m^2^ for this scenario, the prevalence reaches only 50% on day 3, decreases to 6% on the sixth day and further thereafter (Figure 6 and Appendix A). The levels for ESBL in the feces and litter decreased sharply in comparison to the reference, reaching a maximum of only 2.7 ± 3.43 log_10_ CFU/g feces and 1.71 ± 2.45 log_10_ CFU/g in litter, both on day 35 (Table 3).

#### 3.3.4. Combination of Measures

When we model different combinations of management practices, the number of infected birds decreases in all cases, but to different degrees. First, we reduced the stocking density while increasing the litter quantity. In the modeled pen, where one-fifth of the day-old chicks were set positive, the combination litter quantity 3000 g/m^2^ and stocking density of 20 kg/m^2^ (Combination 1) had a stronger effect than when the stocking density was 25 kg/m^2^ (Figure 6, Figure 7 and Figure 8 and Appendix A). With a stocking density of 20 kg/m^2^ and 3000 g litter/m^2^, the prevalence was highest on day 8 at 56.91 ± 28.45% and decreased from day 9 to 21.58 ± 36.05% on day 37 (Figure 6 and Table 3). The excreted bacteria levels reached a maximum of 2.61 ± 2.38 log_10_ CFU/g on day 2, and the litter contained a maximum of 21.58 ± 36.05 log_10_ CFU/g on day 37 (Table 3). In the scenario with the contaminated pen, the uptake and spread of ESBLs does not start at all in this setting (Figure 6). In contrast to the positive chicks, we obtained the same result for the positive pen with a stocking density of 25 kg/m^2^ in combination with 3000 g litter/m^2^ (Appendix A).

By adding the alternative breed to a stocking density of 25 kg/m^2^ and the increased litter (3000 g/m^2^), i.e., Combination 2, we see that there is a strong effect on the colonization of birds with ESBLs. The prevalence is highest on day 6 at 72.03 ± 28%, and the curve drops to 1.83 ± 8.97% on day 37 (Table 3 and Appendix A) in the pen where the birds were set positive on day 1. In the contaminated pen, only 32.12 ± 5.09% of the birds were positive on day 3. From the fourth day, all chicks were negative (Figure 6). The levels of bacteria in feces and litter were low (Figure 7 and Figure 8 and Table 3).

#### 3.3.5. Products with Impact on the Microbiota

Our model also shows that the infection dynamics change drastically when the maximum growth rate of ESBL-producing *E. coli* in the chickens’ intestines is reduced from 10^5^ to 10^4^. In the scenario with the positive birds, the prevalence did not exceed 21%, and both feces and litter contained low levels of bacteria, namely 1.71 ± 1.46 log_10_ CFU/g feces and 0.81 ± 0.56 log_10_ CFU/g litter (Figure 6, Figure 7 and Figure 8). If we model a contaminated pen at the beginning, the effect is different. A prevalence of 77.04 ± 4.79% is reached on day 4, but on the sixth day, the curve drops steeply to 0% (Figure 6). The values for fecal and litter contamination are low, with maxima calculated for the first days only, 1.25 ± 0.63 log_10_ CFU/g feces (day 3) and 1.9 log_10_ CFU/g litter (day 1) (Figure 7 and Figure 8 and Table 3).

## 4. Discussion

Mathematical models, in general, serve to simplify complex problems and, thus, help us to comprehend and study them [55]. Therefore, they are well suited to help us better understand the transmission and spread of AMR [38,56]. Moreover, as in our case, they can replace costly life- and animal-intensive trials. Of course, we also identified limitations. Relevant factors that could have a strong impact on the outcome could not be considered in all details, e.g., environmental influences such as temperature, the competitive behavior of different bacterial species in the gastrointestinal tract or the individual characteristics of organisms, such as growth or inactivation, persistence or resilience [57]. Our model describes the effects of management measures to the colonization of broiler chickens and the dissemination of resistant bacteria within a flock in a pen. The uptake, spread and circulation of ESBL-producing *E. coli* were modeled by using basic assumptions, estimated parameters and data from the literature. Some of these assumptions may underestimate or overestimate the actual processes. For example, the processes in the gastrointestinal tract of broilers are very complex and partly unknown. Therefore, we could only make approximations and work with estimates. However, we were able to show that the model is consistent with observations from experimental studies. For this purpose, we have compared our calculated results with data from the seeder-bird experiments of Dame-Korevaar et al. [43,50] and Robé et al. [49]. After adjusting individual parameters, whose values we initially estimated, we found that the results were very similar: an initial dose of 10^2^ CFU per seeder bird was enough to colonize almost all of the birds in a flock 72 h after setting the seeder birds positive, regardless of the ratio of seeder to sentinels (1:1 or 1:5) and the day of setting them positive, i.e., the equivalent to the day of inoculation. To be precise, our model does not distinguish whether chicks are positive from the first or third day of life, because we did not model the emergence and composition of the gut microbial flora. However, age presumably plays a critical role in colonization, as the development of the microbiota takes place in the first days of chicks’ life [58,59]. In addition to the prevalence data, the excretion levels and the number of CFU in the mixture of litter and feces, obtained by our model, are comparable to the results from experimental studies. For the seeder-bird scenarios without any intervention, the mean excretion is around 4.5 log_10_ CFU/g feces. A wide range of 2 to 9.92 log_10_ CFU/g feces can be found in the literature, due to different designs of the experiments or calculation methods. Laube [60], for example, found an arithmetic mean of 6.89 log_10_ CFU/g feces in pooled fecal samples from three sampling dates from day 1 to day 35 after being housed. Blaak et al. published 10^2^ to 8.3 × 10^9^ log CFU/g, i.e., 2 to 9.92 log_10_ CFU/g, in fresh droppings [54]. In litter, our average values are about 3.75 log_10_ CFU/g. Blaak et al. [54] and Siller et al. [31] published geometric means ranging from 2.98 to 4.72 log_10_ CFU/g litter, with the geometric mean generally lower than the arithmetic mean we used in our model.

We addressed the findings of Daehre et al. [10] and Huijbers et al. [35] and modeled a scenario with a contaminated pen at the beginning of the fattening period. Daehre et al. observed horizontal transmission of ESBL/AmpC-producing *Enterobacteriaceae* from one broiler fattening flock to another, due to a contaminated housing environment after cleaning and disinfection. Huijbers et al. concluded from their studies that the environment, as well as positive day-old chicks, plays an important role in the introduction and transmission of ESBL/AmpC-producing *E. coli* into broiler flocks. Possible sources for the introduction of ESBL-producing *E. coli* into the pen include not only the previous flock but also humans; and companion and wild animals such as birds, rodents or small mammals [61,62,63,64]. For this reason, cleaning and disinfection, as well as pest control and internal hygiene measures, are so important [65,66]. To this end, we investigated different levels of contamination, but found that the number of bacteria below 10^6^ CFU is not sufficient to infect the entire flock during the entire fattening period. Our model calculated a homogenous distribution of this initial contamination and, later, a homogenous distribution of the excretions of ESBL-producing *E. coli* in the litter. Since litter is removed after the fattening period, contamination of a real pen is mainly found in cracks and crevices in floors, walls and ceilings [66,67]. In addition, it can be assumed that litter contamination by excretion is mainly affected at the surface, and survival rates for *E. coli* are, in turn, different at the surface than deeper in the litter, due to temperature and moisture differences [31]. Furthermore, chicks also do not disperse evenly throughout the pen. In the first days, they like to cuddle together or stay mainly in warm spots. Finally, the ratio of feces to litter at the beginning of the fattening period is probably less favorable for bacterial survival than at the end of fattening, when litter may be moist and clumpy. Because of a lower survival rate in the dry litter at the beginning of fattening, the spread of ESBL-producing *E. coli* could be slower. We did not model temperature or humidity, so increasing litter quantity by a factor of three to six did not have much effect in our model with the scenario of positive day-old chicks. Only when we increased the litter amount by a factor of nine (9000 instead of 1000 g/m^2^) was the prevalence strongly reduced. The reason for this could be that the ratio of feces to litter changes little during the fattening period, because the amount of feces excreted in total is high. Even with 9000 g litter/m^2^ in a pen with 90 chicks—the pen area is 5.38 m^2^ (stocking density 39 kg/m^2^, target weight 2.332 kg)—there is 48.6 kg of litter at the beginning instead of 5.4 kg, but, in addition, there is about 450 kg of feces, calculated with 5 kg per chick. It should be noted here that our calculation for excretion and the resulting amount of feces (4–6 kg) are comparable to the literature data of Bolan et al. [68]. In contrast to the scenario just described, increasing the amount of litter in the contaminated pen, has a strong effect. Even an amount of 3000 g/m^2^ litter has a significant impact on the dissemination of ESBL-producing *E. coli*. We explain this by the fact that the amount of ESBL with which we “contaminated” the pen in the model, as mentioned above, was evenly distributed in the litter, so it was highly diluted and no longer sufficient to infect the chickens. In this case, the results of our model differ from the results of the experimental studies of Robé et al. [21], who found a significant increase in the cecum colonization with both resistant *E. coli* strains with the increased litter amount (3000 g/m^2^). In our model, we calculated an impact on the dissemination of resistant bacteria for both scenarios. On the other hand, we agree with the data of Robé et al. [21] on the use of an alternative breed. They found no significant difference in colonization with ESBL- and pAmpC-producing *E. coli*. Our model shows that there is little effect on colonization and transmission of ESBL-producing *E. coli* in either scenario, whether the chicks or the pen is positive at the beginning. However, the difference between the Ross 308 and Rowan x Ranger breeds is probably not just the length of the fattening period and a different amount of feed per day, as included in our model. Possibly the intestinal flora has a different composition, or the immune system reacts differently to bacterial strains. Differences in the microbiota are suspected in different breeds, but there are few published data on this to date [59,69,70].

In our model, we see no effect of reduced stocking density in the scenario where one-fifth of the birds are positive at the beginning of the fattening period. On the other hand, in the initially contaminated pen, a reduced prevalence is already evident with a reduction in stocking density from 39 to 25 kg/m^2^. At a stocking density of 20 kg/m^2^, the influence is even stronger. Presumably, a dilution effect can also be observed here, which is reflected in the ESBL concentrations in the feces and in the litter. At lower stocking densities, chickens do not pick up enough bacteria at the beginning of the fattening period to become infected and start the infection dynamics. It is known that the stocking density in broiler chickens affects the composition of the microbiota [71,72] and colonization of the gastrointestinal tract [73], possibly due to the fact that the stocking density also has an effect on litter moisture [74,75] and, thus, on the survival rate of *E. coli*. Furthermore, the stocking density has an effect on animal welfare, behavior and locomotion [76,77], possibly including pecking and ingestion of potentially contaminated litter material. In their experiments, Robé et al. [21] also found that a lower stocking density reduced the number of ESBL-producing *E. coli* in the cecum of chicks. However, a reduction in pAmpC-producing *E. coli* was not observed.

Feed additives, probiotics or competitive exclusion products could be suitable to have a decisive influence on the multiplication rate in the gastrointestinal tract of broilers and are expected to prevent the colonization of broilers with ESBL-producing *Enterobacteriaceae* or *E. coli* [22,25,78,79]. As we noted in a review [32], the most effective measures seem to be cleaning and disinfection, as well as products influencing the microbiota and thereby suppressing the growth of undesired bacteria in the gastrointestinal tract. In accordance with these findings, our model shows that, in both scenarios, reducing the growth rate in the intestine has a strong impact on the transmission and spread of resistant bacteria. The infection process comes to a standstill.

To the best of our knowledge, there is no study on the effects of a combination of measures on ESBL-producing *E. coli* in broiler farms. Our model suggests that rearing broiler chickens at lower stocking densities, in combination with a higher amount of litter per square meter, reduces the transmission of ESBL-producing *E. coli*. This is in line with expectations for the efficiency of biosecurity measures or a combination of various production parameters, management factors, hygiene measures and further preventive measures. Thus, the control of resistant *E. coli* could benefit from these intervention strategies.

Our model describes the effects of management measures on the colonization of broiler chickens and the dissemination of resistant bacteria within a flock. We were able to use it to simulate numerous virtual experiments and determine the effectiveness of both individual interventions and combinations of management changes in reducing the prevalence of ESBL-producing *E. coli*. However, in part, there are still gaps in our knowledge in the field of AMR and animal husbandry. For example, a more detailed understanding of the effects of stocking density on the uptake rate of resistant bacteria or on growth rates in the gut of chickens could lead to the further improvement of mathematical models such as this one.

## Figures and Tables

**Figure 1 microorganisms-10-00981-f001:**
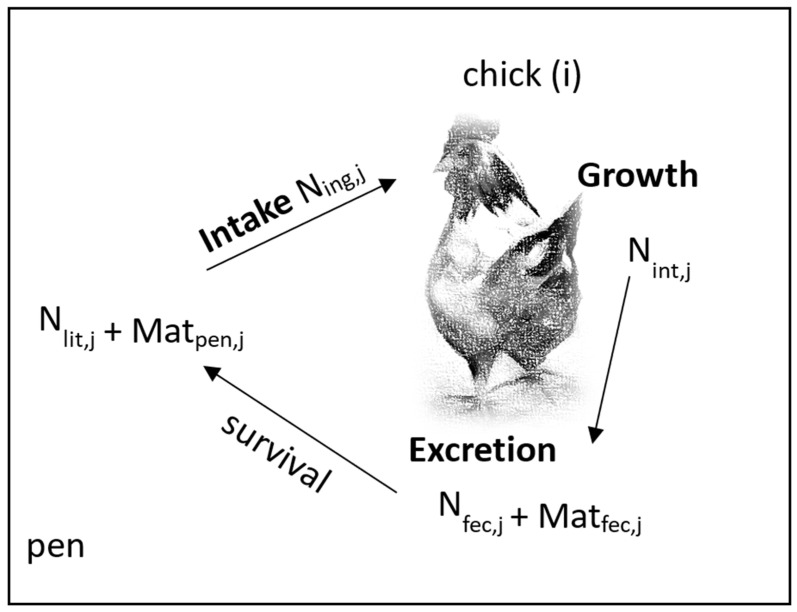
The model calculates the intake (N_ing_), growth (N_int_) and excretion (N_fec_) of ESBL in the pen for each day (j) and chick (i). The survival of excreted ESBL (N_fec_) in the amount of feces (Mat_fec_) is calculated and, thus, so is the contamination of the litter mixed with feces (Mat_pen_). These resistant bacteria present in the pen (N_lit_) can, in turn, be picked up by the chickens.

**Figure 2 microorganisms-10-00981-f002:**
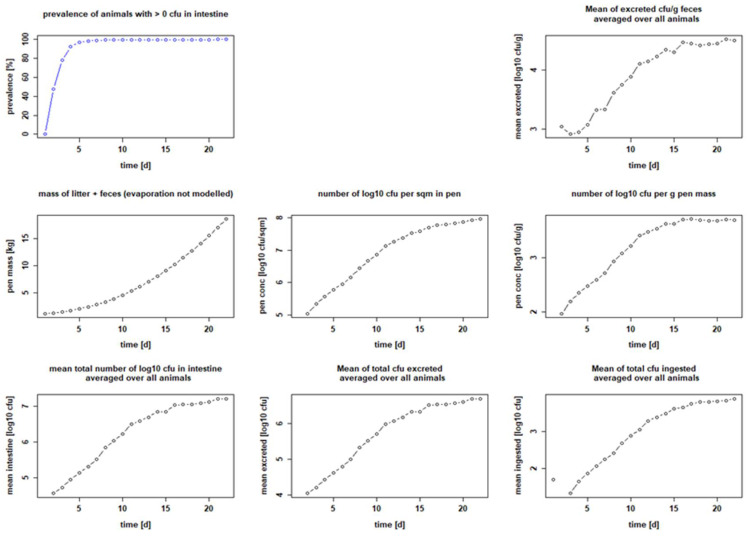
Calculated infection dynamics when seeder birds are set positive with 10^2^ CFU bacteria on day 1. Dotplots with results for seeder-bird experiment with 5 seeders, 5 sentinels, Ross 308, fattening period 21 days, stocking density 20 kg/m^2^ and 1000 g litter/m^2^.

**Figure 3 microorganisms-10-00981-f003:**
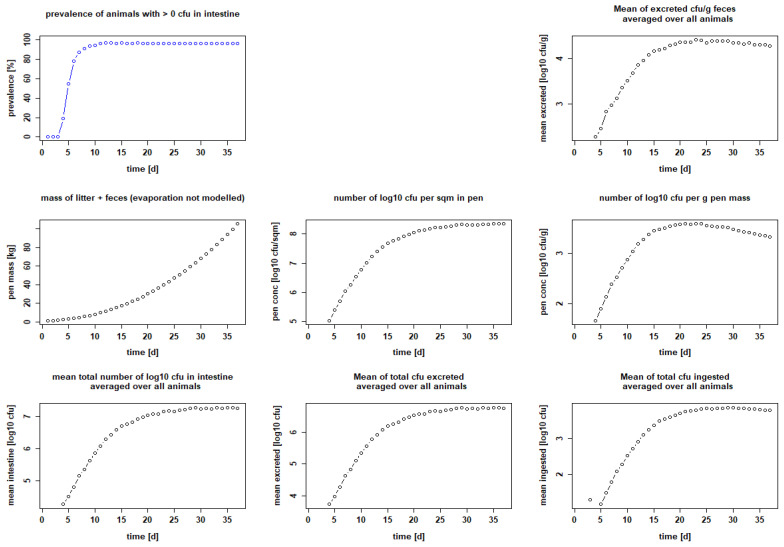
Calculated infection dynamics when seeder birds are set positive with 10^2^ CFU bacteria on day 3. Dotplots with results for seeder-bird experiment with 4 seeders, 16 sentinels, Ross 308, fattening period 36 days, stocking density 39 kg/m^2^, 1000 g litter/m^2^.

**Figure 4 microorganisms-10-00981-f004:**
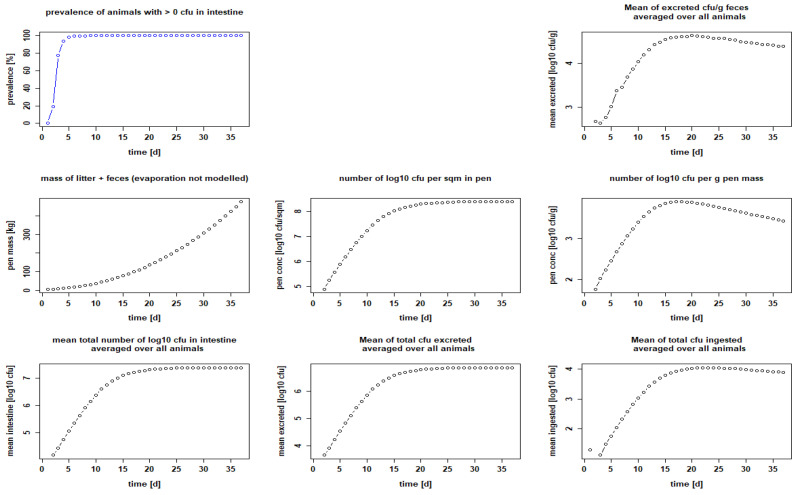
Calculated infection dynamics when chicks are positive (reference). Dotplots with results for seeder-bird experiment with 18 seeder, 72 sentinels, chickens set positive with 10^2^ CFU each on day 1, Ross 308, feeding period 36 days, stocking density 39 kg/m^2^ and 3000 g litter/m^2^.

**Figure 5 microorganisms-10-00981-f005:**
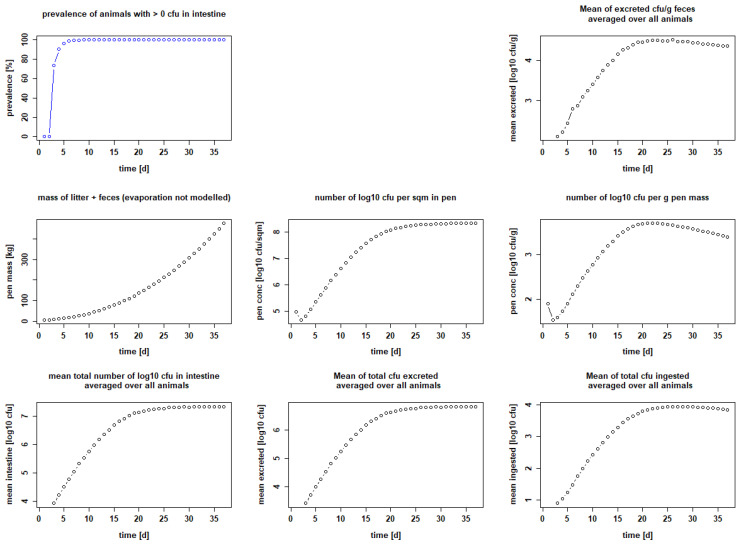
Calculated infection dynamics when the pen is contaminated (reference group). Dotplots with results for 90 chickens, Ross 308, feeding duration 36 days, stocking density 39 kg/m^2^ and 1000 g litter/m^2^; pen contaminated at the beginning with 10^6^ CFU (in 5.38 kg litter).

**Figure 6 microorganisms-10-00981-f006:**
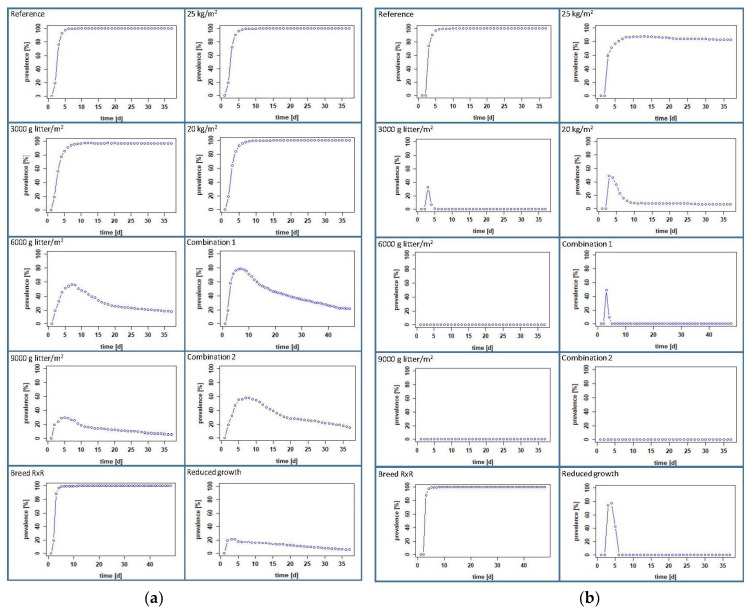
Calculated prevalence of infected birds of the reference groups and the intervention groups through the entire fattening period: (**a**) scenario with positive chicks and (**b**) scenario with a contaminated pen (a positive bird has more than 0 CFU in the intestines).

**Figure 7 microorganisms-10-00981-f007:**
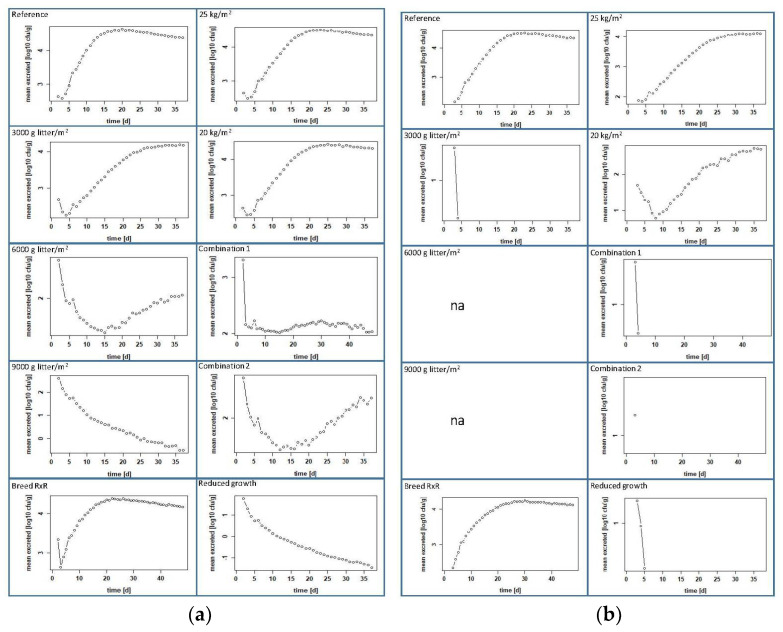
CFU per gram feces (log_10_) of all birds of the reference group and the intervention groups through the entire fattening period: (**a**) scenario with positive chicks and (**b**) scenario with a contaminated pen; “na” means that there are no plottable values.

**Figure 8 microorganisms-10-00981-f008:**
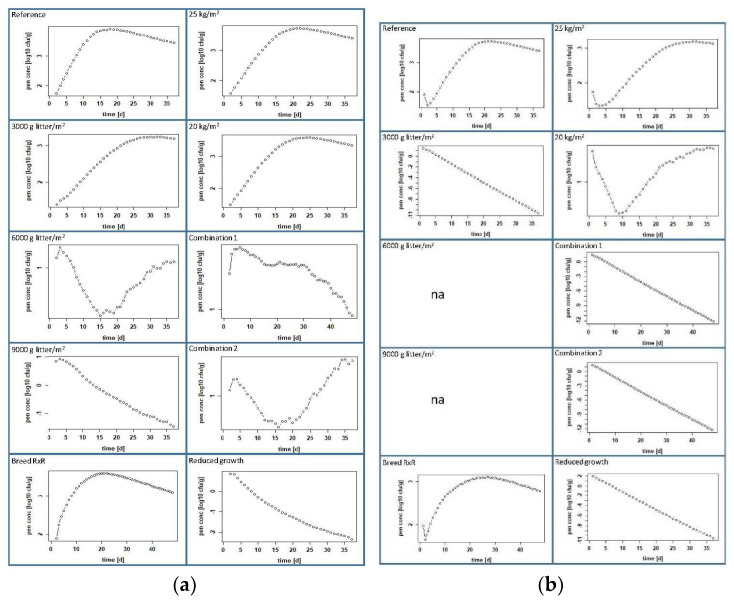
CFU per gram litter (log_10_) of all birds of the control group and the intervention groups through the entire fattening period: (**a**) scenario with positive chicks and (**b**) scenario with a contaminated pen; “na” means that there are no plottable values.

**Table 1 microorganisms-10-00981-t001:** List of the model parameters.

Parameter	Definition	Value and Unit	Reference
Feeding duration	Total duration of the fattening period	36 days (Ross 308) 47 days (Rowan x Ranger)	[21,43]
Feed intake	Daily feed intake for all days of the defined fattening period according to the information provided by the breeder	According to manual [g]	Aviagen manual Ross 308 [44], Table “As-Hatched Performance”, days 2–37; Aviagen manual Rowan Ranger [45], Table “As-Hatched”, days 2–48
Water intake	Daily intake of water for all days of the defined fattening period according to the information provided by the breeder	According to manual [g]	[46]; 70% of the quantities, see manual Aviagen Brief [47]
k_ing_	Litter uptake and intake of ESBLs with contaminated litter	Factor 0.014 (= 1.38% of the amount of feed intake [g])	Estimated [42]
Litter amount	Litter quantity per square meter at the beginning of the fattening period	1000 g/m^2^	Commercial standard in Europe
Stocking density	Maximum number of chickens per square meter (refers to the target weight)	39 kg/m^2^	(Tierschutz-Nutztierhaltungs-verordnung—TierSchNutztV), 2001, § 19, Absatz 7, (3)
k_via_	Random factor for bacteria which do remain capable of reproduction when ingested	0–0.5	Assumed
k_gro_	Random growth rate for ESBL-producing *E. coli* in the chicken’s intestine	10^0^–10^5^	Maximum estimated [48]
Carrying capacity	Maximum number of CFU ESBL-producing *E. coli* in colon	8 log_10_/g	[49,50]
Transition factor	The factor by which the total amount in the chick within 24 h, 1 day, has to be divided to obtain the content of the intestine (in grams)	4.85	Transition time estimated [51,52,53]
k_exc_	Factor of excreted bacteria	0.3	Assumed
k_sur_	Daily survival rate of ESBL in litter	0.5	Estimated [31,54]
Target weight	Weight of the chickens at the end of the fattening period (see “feeding duration” above)	2.332 kg (Ross 308) 1.911 kg (Rowan x Ranger)	Aviagen manual Ross308-308FF-Broiler [44], Table “As-Hatched Performance”. Aviagen manual Rowan Ranger [45] (Appendix 3, Table 1), “As-hatched broiler performance”.

**Table 2 microorganisms-10-00981-t002:** Calculated prevalence and bacterial counts for the simulated scenarios with seeder birds introducing ESBL-producing *E. coli* into the flock (100 iterations).

Seeder–Sentinel Ratio	No. of Chicks	Feeding Duration [Days]	Inoculation		Prevalence * [%]		ESBL-Producing *E. coli* in Feces * [log_10_ CFU/g]		ESBL-Producing *E. coli* in Litter * [log_10_ CFU/g]	
			Amount	On Day	Maximum	On Day	Max.	On Day	Max.	On Day
1:1	10	21	10^2^ CFU	1	99.2 ± 4.42	7	4.51 ± 4.42	21	3.71 ± 3.66	17
1:5	20	36	10^2^ CFU	3	97.15 ± 14.67	15	4.41 ± 4.26	23	3.6 ± 3.43	23

* Both the average of the prevalence and the CFU values per gram litter are calculated over all iterations. The excreted CFU values are obtained by calculating the average over all animals in one iteration at first and then over all iterations.

**Table 3 microorganisms-10-00981-t003:** Calculated prevalence and bacterial counts for the control groups and the simulated management measures on flocks with 90 chickens (100 iterations).

Scenario	Feeding Duration [Days]	Prevalence * [%]		ESBL-Producing *E. coli* in Feces * [log_10_ CFU/g]		ESBL-Producing *E. coli* in Litter * [log_10_ CFU/g]	
		Maximum	On Day	Maximum	On Day	Maximum	On Day
**Reference groups**							
Chicks positive	36	99.38 ± 1.98	6	4.63 ± 4.08	20	3.9 ± 3.47	18
Pen positive	36	99.79 ± 0.61	7	4.51 ± 3.85	26	3.68 ± 3.36	20
**Management measure 3000 g litter/m^2^**							
Chicks positive	36	98.53 ± 6.24	11	4.19 ± 3.92	34	3.25 ± 3.04	30
Pen positive	36	32.46 ± 4.42	3	1.48 ± 1.14	3	1.47 ± Inf	1
**Management measure 6000 g litter/m^2^**							
Chicks positive	36	57.23 ± 25.11	7	2.63 ± 2.37	2	1.13 ± 0.77	3
Pen positive	36	0	0	0	0	1.18 ± Inf	1
**Management measure 9000 g litter/m^2^**							
Chicks positive	36	29.72 ± 16.97	5	2.64 ± 2.43	2	0.94 ± 0.59	3
Pen positive	36	0	0	0	0	1.01 ± Inf	1
**Management measure slow growing breed**							
Chicks positive	47	99.59 ± 1.82	6	4.23 ± 4.01	25	3.4 ± 3.33	21
Pen positive	47	99.19 ± 1	5	4.01 ± 3.7	30	3.11 ± 2.84	27
**Management measure stocking density 25 kg/m^2^**							
Chicks positive	36	99.16 ± 4.15	7	4.5 ± 4.04	22	3.7 ± 3.29	21
Pen positive	36	90.64 ± 28.71	16	4.1 ± 3.98	30	3.2 ± 3.04	32
**Management measure stocking density 20 kg/m^2^**							
Chicks positive	36	99.04 ± 5.68	7	4.44 ± 3.92	26	3.59 ± 3.15	23
Pen positive	36	49.28 ± 5.21	3	2.7 ± 3.43	35	1.71 ± 2.45	35
**Combination of measures litter 3000 g/m^2^ and stocking density 20 kg/m^2^ (Combination 1)**							
Chicks positive	36	56.91 ± 28.45	8	2.61 ± 2.38	2	1.47 ± 1.93	37
Pen positive	36	0	0	0	0	1.19 ± Inf	1
**Combination of measures breed RxR, litter 3000 g/m^2^ and stocking density 25 kg/m^2^ (Combination 2)**							
Chicks positive	47	72.03 ± 28	6	3.3 ± 3.07	2	1.41 ± 1.14	4
Pen positive	47	32.12 ± 5.09	3	1.29 ± 0.95	3	1.37 ± Inf	1
**Measure on microbiota: reduced growth of ESBL *E. coli* in the intestine**							
Chicks positive	36	20.68 ± 6.97	3	1.71 ± 1.46	2	0.81 ± 0.56	2
Pen positive	36	77.04 ± 4.79	4	1.25 ± 0.63	3	1.9 ± Inf	1

* Both the average of the prevalence and the CFU values per gram litter are calculated over all iterations. The excreted CFU values are obtained by calculating the average over all animals in one iteration at first and then over all iterations. The decadic logarithm of zero or negative values is negative infinite (-Inf).

## Data Availability

Not applicable.

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
