# Peer review of "Modeling the Impact of Management Changes on the Infection Dynamics of Extended-Spectrum Beta-Lactamase-Producing Escherichia coli in the Broiler Production"

_microorganisms, 2022, doi:10.3390/microorganisms10050981_

Round 1

Reviewer 1 Report

I would like to congratulate the authors very interesting and well-written paper! The publication raises a current and relevant scientific issue. Tools using mathematical modeling can be of great help to the poultry industry as well as scientists. The materials and methods as well as the results were described very carefully and precisely.

Author Response

Thank you very much for your positive feedback on our manuscript.

Reviewer 2 Report

Comments to the authors

  • L42: use the term ‘’AMR’’ instead of ‘’antimicrobial resistance’’
  • L46: Escherichia coli (coli)
  • L92: correct the sentence
  • L136: colony-forming unit (CFU)
  • L207: remove the full stop
  • L273: remove the full stop
  • L123: coli – use italics
  • L257: add the correct supplementary file
  • L537-540: rewrite the sentence
  • L542: correct the reference as Robé et al. [49]
  • L460: use the term ‘’AMR’’ instead of ‘’antimicrobial resistance’’
  • L581-582: use the term ‘’AMR’’

Author Response

Thank you for your feedback and the comments made on the manuscript.

L42: use the term “AMR” instead of “antimicrobial resistance”

L43: we accepted your suggestion and changed the wording

L46: Escherichia coli (E. coli)

L46: we have introduced the abbreviation in L14

L92: correct the sentence

L92-93: the cross-reference was set incorrectly, but is also dispensable, so we have deleted it

L123: coli – use italics –

L123: we have checked all bacteria names and put them in italics

L136: colony-forming unit (CFU)

L136: we accepted your suggestion and introduced the abbreviation in L131

L207: remove the full stop

L208: we have removed the full stop

L257: add the correct supplementary file

L258: we have added the supplementary Table numbers

L273: remove the full stop

L274: we have removed the full stop

L460: use the term “AMR”

L461: we changed the wording

L537-540: rewrite the sentence

L542-544: we have changed the wording

L542: correct the reference as Robé at al. [49]

L539: we added the reference [21] which is the correct one

L581-582: use the term “AMR”

L583: we have changed the wording

Reviewer 3 Report

Comments to authors:

Overview and general recommendation:

Overall, the study is well designed. All the experiments are accurately performed. The results are clearly presented and highly interpreted. Although the flaws within the manuscript, I suggest its publication in case of minor revision.
Some indications for minor revisions are given below.

Try to develop the introduction

Try to add a conclusion section with future outlook of such mathematical models and their efficiency and/or feasibility in other systems.

Line 479: "To be precise, our model does not distinguish whether chicks are positive from the first or third day of life". What do you think about the accuracy of the studied mathematical model?

Check all the text in order to put commas and punctuation in right places to give the meaning to the sentences and ideas.

Check italic mode all through the text (name of bacteria, etc...)

The style of English language should be verified.

Author Response

We thank the reviewer for the reading and the commenting on our manuscript.

Try to develop the introduction. – We have revised the introduction and hopefully optimized it in your sense.

Try to add a conclusion section with future outlook of such mathematical models and their efficiency and/or feasibility in other systems. - Thank you for the suggestions, which we believe we are addressing in the discussion. Our model is just one piece of the puzzle, which hopefully fits into the big picture and contributes to the further development and optimization of mathematical modeling.

Line 479: “To be precise, our model does not distinguish whether chicks are positive from the first or third day of life”. What do you think about the accuracy of the studies mathematical model?

L480-482: We have changed the wording to improve the statement and make it more understandable.

Check all the text in order to put commas and punctuation in right places to give the meaning to the sentences and ideas. . - We have revised the manuscript, checked and improved the grammar.

Check italic mode all through the text (name of bacteria, etc…) . - We have checked the manuscript and corrected the italic mode where necessary.

The style of English language should be verified.  - We have checked the manuscript and improved the language where necessary.